# Xanthones: Biosynthesis and Trafficking in Plants, Fungi and Lichens

**DOI:** 10.3390/plants12040694

**Published:** 2023-02-04

**Authors:** Camilla Badiali, Valerio Petruccelli, Elisa Brasili, Gabriella Pasqua

**Affiliations:** Department of Environmental Biology, Sapienza University of Rome, P.le Aldo Moro 5, 00185 Rome, Italy

**Keywords:** xanthones, biosynthetic pathways, plants, fungi, lichens, subcellular and cellular localization, biological activity

## Abstract

Xanthones are a class of secondary metabolites produced by plant organisms. They are characterized by a wide structural variety and numerous biological activities that make them valuable metabolites for use in the pharmaceutical field. This review shows the current knowledge of the xanthone biosynthetic pathway with a focus on the precursors and the enzymes involved, as well as on the cellular and organ localization of xanthones in plants. Xanthone biosynthesis in plants involves the shikimate and the acetate pathways which originate in plastids and endoplasmic reticulum, respectively. The pathway continues following three alternative routes, two phenylalanine-dependent and one phenylalanine-independent. All three routes lead to the biosynthesis of 2,3′,4,6-tetrahydroxybenzophenone, which is the central intermediate. Unlike plants, the xanthone core in fungi and lichens is wholly derived from polyketide. Although organs and tissues synthesizing and accumulating xanthones are known in plants, no information is yet available on their subcellular and cellular localization in fungi and lichens. This review highlights the studies published to date on xanthone biosynthesis and trafficking in plant organisms, from which it emerges that the mechanisms underlying their synthesis need to be further investigated in order to exploit them for application purposes.

## 1. Introduction

The designation “xanthone” derives from the Greek word “xanthós”, meaning yellow, and was coined by Schmid in 1855 to indicate the yellow color of the compound isolated from the pericarp of mangosteen (*Garcinia mangostana* L.), a tropical fruit belonging to the Clusiaceae (or Guttiferae) family.

Xanthone is an aromatic oxygenated heterocyclic molecule, with a dibenzo-γ-pirone scaffold, known as 9H-xanthen-9-one, with the molecular formula of C_13_H_8_O_2_ (Figure 1). The number and class of rings A and B derive from the biosynthetic pathways in higher plants leading to the acetate-derived A-ring (carbons 1–4) and the shikimic acid pathway-derived B-ring (carbons 5–8). Xanthone derivatives consist of slight differences that can be found depending on the nature of the substituents and their localization on the scaffold. Different possible configurations of the two benzene rings and various substituents can be found, leading to higher complexity. 

Xanthones are widely distributed in nature and synthesized by several living organisms, even when phylogenetically distant from each other. By January 2016, the *Dictionary of Natural Products* revealed that natural xanthones are ca. 2000, including their reduced derivatives di-, tetra- and hexahydroxanthones. Plants remain the prevalent source of xanthones, accounting for nearly 80% of natural xanthones. In a pie diagram, shown by Pierre Le Pogam and Joël Boustie, fungi represent 15% while lichens represent the remaining 5% [1]. Algae and bacteria are also able to synthesize xanthones [2,3,4]. Xanthones have been found even in fossil fuels, demonstrating their chemical stability [3,5].

Historically, the first natural xanthone described was gentisin (1,7-dihydroxy-3-methoxyxanthone) isolated from the plant *Gentiana lutea* L. in 1821 [6] and the first prenylxanthone derivative, tajixanthone, was isolated from the mycelium of *Aspergillus stellatus* in 1970 [7]. The most studied plant species producing xanthones belong to the Clusiaceae, Hypericaceae, Gentianaceae and Cariophyllaceae families [8,9,10]. Xanthone is well-known to have “privileged structures” because this simple tricyclic compound exhibits wide biological activities such as anticancer, antimicrobial, antifungal, antimalarial, anti-HIV, anticonvulsant, anticholinesterase, antioxidant, anti-inflammatory, and antimalarial activities, among others [11,12,13,14,15,16,17,18,19]. Their interesting structural scaffold and pharmacological importance have encouraged scientists to isolate these compounds from natural products and synthesize them as novel drug candidates. For this reason, biotechnological strategies for xanthone production, such as cell and root cultures grown in bioreactors, and hairy roots, have been studied in recent years in order to obtain efficient protocols for future large-scale production [15,20,21,22,23,24,25,26,27,28].

In view of the importance of xanthone derivatives in medicinal chemistry, we have made efforts to summarize the different classes of xanthones, their biosynthesis and trafficking mechanisms in plant organisms reported in the literature over the last decades. 

This review shows the current knowledge of the xanthone biosynthetic pathway with a focus on the precursors and the enzymes involved, as well as on the cellular, tissue and organ localization of xanthones in plants. As far as fungi and lichens are concerned, only biosynthetic pathways have been studied; no information is yet available on the subcellular and cellular localization in these organisms. The selected articles about the topic xanthone biosynthesis, and trafficking in plant organisms, have been screened from Web of Science, Scopus, Pubmed, and Google Scholar to highlight the current advancement and future direction toward completion of the biosynthetic pathways of xanthones.

## 2. Classification

In recent decades, there has been widespread interest in studying the classification of xanthones, motivated primarily by the great potential of these compounds for their medically useful biological properties. Because of the great diversity of substituents and the discovery and synthesis of new xanthones, their classification by groups has evolved. With the exception of simple xanthones that have only methyl groups attached to the core structure, all other xanthones can be divided into six main groups based on their substituents: oxygenated xanthones, glycosylated xanthones, prenylated xanthones, xantholignoids, bisxanthones, and various xanthones [29,30]. These molecules are biosynthesized and accumulated in various plant organs (leaves, stems, roots, flowers, and fruits) and tissues, and in other organisms [3,31,32,33]. The xanthones identified in plants, fungi, and lichens are reported in Table 1, Table 2 and Table 3.

### 2.1. Oxygenated Xanthones

Oxygenated simple xanthones are xanthones with simple substituents such as hydroxyl or methoxy, subdivided according to the degree of oxygenation into unoxygenated, mono-, di-, tri-, tetra-, penta-, and hexaoxygenated molecules [232,233]. This class of xanthones is produced both by plants, but also by fungi and lichens (Table 1, Table 2 and Table 3). This group is abundant in many natural products and is the starting point for many more complex xanthones [232,234]. Many monooxygenated xanthones have been isolated from the Gentianaceae family, for example, 2-hydroxyxanthone (Figure 2), 4-hydroxyxanthone and 2-methoxanthone; however, these xanthones have also been found in other plant families such as Clusiaceae, Hypericaceae, Moraceae (Table 1).

More than fifteen deoxygenated xanthones have been reported from plants in the family Clusiaceae. 1,5-dihydroxyanthone, 1,7-dihydroxyanthone, and 2,6-dihydroxyanthone have been found quite widely. Other deoxygenated xanthones such as 1-hydroxy-5-methoxanthone, 1-hydroxy-7-methoxanthone, 2-hydroxy-1-methoxanthone, 3-hydroxy-2-methoxanthone, 3-hydroxy-4-methoxanthone, 5-hydroxy-1-methoxanthone, and 1,2-methylenedioxanthone have been reported from plants belonging to Polygalaceae and Calophyllaceae families (Table 1) [97,193]. The natural trioxygenated xanthones 1,3-dihydroxy-5-methoxy-xanthone-4-sulfonate and 5-O-β-D-glucopyranosyl-1,3-dihydroxy-xanthone-4-sulfonate were isolated from *Hypericum sampsoni* Hance., and *Centaurium erythraea* [137,176]. Tetraoxygenated xanthones have been reported mainly from plants in the families Gentianaceae, Clusiaceae, and Polygalaceae. The xanthones isolated from *Polygala vulgaris* L. were 7-chloro-1,2,3-trihydroxy-6-methoxanthone, 1,3,5,6-, 1,3,5,7-, and 1,3,6,7-tetrahydroxanthones [196].

Xanthones isolated from *Gentiana rhodantha* Franch. were methylated pentaoxygenated xanthones, namely 1,8-dihydroxy-2,3,7-trimethoxanthone, 5,6-dihydroxy-1,3,7-trimethoxanthone, 1,7-dihydroxy-2,3,8-trimethoxanthone, and 3,8-dihydroxy-1,2,6-trimethoxanthone [147]. In addition, hexoxygenated xanthones such as 8-hydroxy-1,2,3,4,6-pentamethoxanthone, 1,8-dihydroxy-2,3,4,6-tetramethoxanthone and 1-hydroxy-3,5,6,7,8-pentamethoxyxanthone have been isolated from two species of *Centaurium* [4,137,138] (Table 1).

### 2.2. Xanthone Glycosides

Xanthone glycosides are synthesized in higher plants and mainly found in the families Gentianaceae and Polygalaceae (Table 1). These xanthones are known to have a glycosidic residue associated with a C- or O-. Many xanthones of this group have the glycosidic residue linked to C-6 carbon and may consist of xylose, glucose, or epiose. The presence of two glycosidic residues can also be observed, the second generally being linked to the C-2 of the main structure [235]. Mangiferin and isomangiferin are the most common C-glycosides (Figure 3a). Mangiferin (2,-C-β-Dglucopyranosyl-1,3,6,7-tetrahydroxyxanthone) is of widespread occurrence in angiosperms and ferns and was first isolated from Mangifera indica [236]. Natural xanthone O-glycosides are restricted to the family Gentianaceae and in this review gentiacauloside from *Gentiana acaulis* L., gentioside from *G. lutea*, and swertianolin (Figure 3b) from *Swertia japonica* Makino have been reported [143,144,156].

### 2.3. Prenylated Xanthones

There is considerable variability in the classification of prenylated xanthones; in fact, it is possible to observe the presence of the prenyl group in different positions of the basic structure. These groups can have carbon chains consisting of a single prenyl group, composed of five carbons (e.g., allanxanthone A, Figure 4), or multiples such as 10 or 15 carbons. In addition, prenylated xanthones can have varying degrees of oxygenation. These molecules are predominantly extracted from species belonging to Hypericaceae family, such as *H. sampsonii* (Table 1), from which an antiviral xanthone against hepatitis B, hyperxanthone, has been isolated [176]. Numerous prenylated xanthones have also been isolated from fungi belonging to the genera *Actinomadura, Emericella* and *Paecilomyces* [200,201,204,212]. Other prenylated xanthones isolated from fungi are reported in Table 2.

### 2.4. Bisxanthones

Bisxanthones were identified in higher plants, fungi, and lichens [3,165], and reported in Table 1, Table 2 and Table 3. Their chemical structure consists of a 9H-xanthen-9-one dimer with several substituent groups. These include jacarelhyperols A and B isolated from the aerial parts of *Hypericum japonicum Thunb* and the dimeric xanthone, and globulixanthone E (Figure 5), from the roots of *Symphonia globulifera* L.f. [132,133,165]. Many xanthones belonging to this group have also been isolated in lichens, particularly in the genera *Teloschistale* and *Diploicia* (Table 3). In addition, tetrahydroxanthone dimeric C2-C2 dicerandrols A, B, and C have been isolated from the fungus *Phomopsis longicolla* [237] (Table 2). A dimeric C4-C4 xanthone isolated from the root bark of *Centaurium erythrea* Raf. has remarkable antimicrobial properties against gram positives such as *Staphylococcus* spp. [139].

### 2.5. Xantholignoids

The group of xantholignoids, characterized by a connection between xanthones and lignin (conifer alcohol) structures, is limited. Cadensins A and B were isolated from *Caraipa densifora* Kubitzki. Kielcorin (Figure 6) was initially obtained from *Hypericum* species but has also been isolated from *Vismia guaramirangae* Huber [181], *Kielmeyera variabilis* Mart. & Zucc. [99], and *Hypericum canariensis* L. (Table 1), [163]. In fungi and lichens, this class of xanthones is present in many genera (Table 2 and Table 3).

### 2.6. Miscellaneous Xanthones

Miscellaneous xanthones are defined for all xanthone derivatives that cannot be classified into other groups. They are found in the kingdom of plants and fungi (Table 1, Table 2 and Table 3). Among them, we include xanthofulvin (Figure 7) and vinaxanthone, isolated from *Penicillium* spp. (Table 1) SPF-3059 [238], and thioxanthones and azaxanthones [239].

## 3. Xanthone Biosynthesis

### 3.1. Xanthone Biosynthesis in Plants

#### 3.1.1. Shikimate Pathway

Xanthones are synthesized in plants via the shikimate pathway with the contribution of the acetate (or polyketide) pathway. Shikimate links carbohydrate metabolism, glycolysis and pentose phosphate pathway, to aromatic compound biosynthesis (Figure 8). The shikimate pathway occurs in green and non-green plastids, thus dependently or independently from light [240]. However, it is known that non-photosynthetic tissues are partially supplied with amino acids transported by the phloem, so production does not occur exclusively within the cell; it can also occur in other tissues or organs, and then transport to other locations occurs [241]. Moon and Mitra [149] showed that shikimate dehydrogenase (SKD) and shikimate kinase (SK), key enzymes of the shikimate pathway, are activated after elicitation by a Ca^2+^-mediated H_2_O_2_ generation, leading to a consequent increase in the xanthone biosynthesis, giving further confirmation to the role of xanthones as defense metabolites as described by numerous articles on the subject [242]. This study revealed for the first time the link between ROS and the pathways involved in xanthone biosynthesis.

After the shikimate pathway, xanthone biosynthesis can proceed with an L-phenylalanine-dependent pathway, as in *Hypericum androsaemum* L. [243], *G. mangostana*, and *G. lutea* [145,244] or an L-phenylalanine-independent pathway, as in *Swertia chirata* Buch.-Ham. ex Wall. [151], *C. erythraea* [243,245,246] and *Hoppea fastigiata* Griseb. [151]. Both the phenylalanine-dependent and phenylalanine-independent pathways pass through the production of 2,3′,4,6-tetrahydroxybenzophenone (2,3′,4,6-THB), which is therefore a central intermediate in the biosynthesis of xanthones (Figure 8).

#### 3.1.2. Phenylalanine-Dependent Pathway

In the phenylalanine-dependent pathway, shikimate forms the amino acid phenylalanine through numerous reactions occurring in two different cell compartments, plastid and cytosol [247,248] (Figure 8). Phenylalanine is biosynthesized from chorismate, the final product of the shikimate pathway. In plastids, chorismate is converted to prephenate which in turn is transaminated producing arogenate. This compound is then dehydrated/decarboxylated to phenylalanine which is then transported to cytosol by the plastidial cationic amino acid transporter (pCAT) [249]. In plants, the arogenate pathway is the predominant route for phenylalanine biosynthesis although another pathway, more common in microorganisms [250], has been described. This route, which has yet to be clarified, involves phenylpyruvate, another product downstream of prephenate. Phenylpyruvate may originate from prephenate in plastids by the action of arogenate dehydratases (ADTs) [251,252] or in the cytosol, requiring a cytosolic pool of prephenate supposedly formed by the action of a cytosolic chorismate mutase (CM) from chorismate previously synthesized in the plastid and then transported to the cytosol [249]. Indeed, as described by Yoo and co-workers [253] in *Petunia hybrida* E.Vilm, prephenate seems to be produced in the plastid but converted to phenylalanine in the cytosol by a phenylpyruvate aminotransferase (PPY-AT), which preferentially uses prephenate as a substrate, suggesting that this alternative route to phenylalanine biosynthesis is also active in the plants. Once in the cytosol, the amino acid is converted to trans-cinnamic acid by the action of the enzyme phenylalanine ammonia-lyase (PAL), which catalyzes the deamination. Trans-cinnamic acid is the substrate of cinnamate-CoA ligase (CNL), which leads to cinnamoyl-CoA. Cinnamoyl-CoA is an intermediate from which benzoyl-CoA is formed as a result of three reactions that are catalyzed by the enzymes cinnamoyl-CoA hydratase/lyase (CHL), benzaldehyde dehydrogenase (BD), and benzoate-CoA ligase (BZL). BZL expression has been demonstrated to increase before xanthone biosynthesis, when the plant is exposed to elicitation, suggesting its role in the biosynthetic pathway upstream of xanthones. Singh and co-workers [254] have shown in *Hypericum calycinum* L. that BZL is localized in both peroxisomes and cytosol, indicating the activation of the CoA-dependent non-β-oxidative pathway for benzoyl-CoA production. The activation of this pathway was previously demonstrated at the biochemical level in *Hypericum androsaemum* L. cell cultures [255]. Furthermore, it is hypothesized that the enzyme is purely involved in the phenylalanine-dependent pathway having benzoic acid as a preferential substrate.

The subsequent reaction is catalyzed by benzophenone synthase (BPS), a type III polyketide synthase, which condenses the benzoyl-CoA molecule with three malonyl-CoAs originating 2,4,6-trihydroxybenzophenone (2,4,6-THB). BPS in *H. androsaemum* and *G. mangostana* has benzoyl-CoA as a specific substrate, suggesting that the phenylalanine-dependent pathway is the one followed for xanthone production in these species [243]. CYP81AA, a cytochrome P450 (CYP) monooxygenase that possesses benzophenone 3′-hydroxylase (B3′H) activity, converts 2,4,6-THB to 2,3′,4,6-THB. Thus, these compounds are the precursors of various benzophenones and xanthones. The two main precursors of xanthones are formed from 2,3′,4,6-THB ring closure. 1,3,5-trihydroxyxanthone (1,3,5-THX) and 1,3,7-trihydroxyxanthone (1,3,7-THX) originate from oxidative phenol coupling reaction that occurs either at the ortho or para position of the 3′-OH group, respectively. Cyclization to 1,3,5-THX and 1,3,7-THX depends on the species [146,256,257,258]. These reactions are catalyzed by two xanthone synthases belonging to the CYP oxidases [256]. They are now known as 1,3,5-THX synthase (CYP81AA2) and 1,3,7-THX synthase (CYP81AA1), respectively [258,259]. One of the two pathways could be used preferentially by a species, but it has been shown from transcriptome databases of *Hypericum* spp. that genes for both CYPs are present, so both isomers of the enzyme could be synthesized in a species in response to certain signals [145,258]. Kitanov and Nedialkov [260] proposed that 1,3,7-THX is generated from 2,4,5′,6-tetrahydroxybenzophenone-2′-O-glucoside (hypericophenonoside) in *H. annulatum* firstly removing the glucoside group by hydrolysis before cyclization. Many different xanthones will then be produced from these precursors, although to date the biosynthetic pathways of many of them are only assumed.

#### 3.1.3. Phenylalanine-Independent Pathway

In the phenylalanine-independent pathway, the biosynthetic pathway originates from shikimate to produce 3-hydroxybenzoic acid in the cytoplasm without the involvement of phenylalanine (Figure 8). To date, how shikimate leaves plastids and which enzymes are responsible for the conversion to 3-hydroxybenzoic acid is unknown. The 3-hydroxybenzoic acid is then thioesterified by 3-hydroxybenzoateCoA ligase (3-BZL) to form 3-hydroxybenzoyl-CoA, and subsequent condensation by BPS leads to the formation of 2,3′,4,6-THB. In *Centaurium* species, 3-BZL enzyme has been shown to have 3-hydrobenzoic acid rather than benzoic acid as a preferred substrate, suggesting that the phenylalanine-independent pathway is the one followed in these species [261]. The biosynthetic pathway continues as described for the phenylalanine-dependent route.

Although the phenylalanine-dependent pathway is more studied and it is assumed that most xanthones are produced downstream of phenylalanine or indistinctly by both the phenylalanine-dependent and independent pathways, some xanthones such as 1,3,5,8-tetrahydroxy xanthone and 1,5,7-trihydroxy-3-methoxy xanthone appear to be produced only through the phenylalanine-independent pathway [151] (Figure 8).

#### 3.1.4. Xanthone Derivatives of 1,3,5-Trihydroxyxanthone

Xanthone-6-hydroxylases (X6H), a CYP-dependent monooxygenase, has been shown to hydroxylate 1,3,5-THX to 1,3,5,6-tetrahydroxyxanthone (1,3,5,6-THX) in *H. androsaemum* and *C. erythraea* [262]. In *S. chirata*, the hydroxylation of 1,3,5-THX occurs at the C-8 position of the ring, originating 1,3,5,8-tetrahydroxyxanthone (1,3,5,8-THX) [263]. On the other hand, Beerhues and Berger [245], studied the elicited cell cultures of *C. erythraea* and *C. littorale* proposing a direct formation of 1,5-dihydroxy-3-methoxyxanthone from 1,3,5-THX. Moreover, the authors proposed a biosynthetic pathway downstream 1,3,5-THX in the cell cultures of these species which produce xanthones such as 1,5-dihydroxy-3-methoxyxanthone, 1-hydroxy-3,5,6,7tetramethoxyxanthone, and 1,8-dihydroxy-3,5-dimethoxyxanthone. However, the enzymes involved in these reactions have not been identified [10,245].

#### 3.1.5. Xanthone Derivatives of 1,3,7-Trihydroxyxanthone

Many more xanthones derive from 1,3,7-THX. X6H is also involved in the formation of 1,3,7-THX derivatives [264,265]. Indeed, the hydroxylation of 1,3,7-THX forms 1,3,6,7-tetrahydroxyxanthone (1,3,6,7-THX) in *H. androsaemum* and *G. mangostana* [262] and potentially resides in the endoplasmic reticulum [258].

1,3,7-THX is proposed to be a precursor compound for prenylated xanthones, such as rubraxanthone from *Garcinia* [266,267] and *Calophyllum* species [9,268,269] and scortechinone B from *Garcinia scortechinii* King [270,271], as well as simple xanthones, such as 1,7-dihydroxy-3-methoxyxanthone (gentisin) and 1,3-dihydroxy-7-methoxyxanthone (isogentisin) from *G. lutea* [146,272]. In *G. mangostana*, γ-mangostin is proposed to be generated by prenylation of the 1,3,6,7-tetrahydroxyxanthones, and α-mangostin by the subsequent O-methylation [273,274]. Another pathway which produces patulone, hyperxanthone E, and hyperixanthone A starting from 1,3,6,7tetrahydroxyxanthones has been reported in *Hypericum* spp. [275,276]. Two enzymes involved in these reactions are known: 8-prenylxanthone-forming prenyltransferase (PT8PX) and patulone-forming prenyltransferase (PTpat). The former has prenylation activity and is mainly localized at the envelope of the chloroplast [276] (Figure 8). The latter is also a prenyltransferase which prenylates the reaction product of the previous reaction, 8-prenyl-1,3,6,7tetrahydroxyxanthone, and produces patulone [275,276]. Other xanthones are supposed to be formed from this route, such as hyperxanthone A and E, but the enzymes involved are unknown [275,276].

Among the glycosylated xanthones, norathyriol 6-O-glucoside (tripteroside or Xt1) and norathyriol-6-O-(6′-Omalonyl)-glucoside (Xt2) have recently been characterized at the molecular level [33]. The enzymes responsible for the reaction that produces these xanthones from 1,3,6,7-THX are norathyriol 6-O-glucosyltransferase (StrGT9) and malonyl-CoA acyltransferase (StrAT2). StrGT9 glucosylates 1,3,6,7-THX to Xt1, which in turn is malonylated in the presence of malonyl-CoA to Xt2 by StrAT2 (Figure 8).

Mangiferin is a well-studied C-glucoside xanthone. A route for its biosynthesis was proposed by Fujita and Inoue [40] and Chen and co-workers [277] in *Anemarrhena asphodeloides* Bunge and *M. indica*, respectively, and reviewed by Ehianeta and co-workers [278]. The results suggest that mangiferin and related xanthone C-glycosides are produced through an intermediate, maclurin 3-C-glucoside, which is converted to mangiferin and isomangiferin by C-glycosyltransferase (CGT).

### 3.2. Xanthone Biosynthesis in Fungi and Lichens

Xanthones are highly unique in fungi and lichens, legitimating, compared to plants, the vastness of chemical diversity of these “privileged structures” with a pronounced biological activity. The biosynthesis of xanthones in fungi and lichens is a topic of considerable interest, but up to now only a few studies report detailed information on the metabolic pathways and trafficking in these organisms.

Xanthones from fungi result from biosynthetically distinct pathways compared to those in plants. The synthesis of the xanthones in fungi has been suggested in 1953 by Birch and Donovan [279], and studied in detail in 1961 by Roberts through the radiolabeled acetate feeding experiments [280] that showed polyketides are the biosynthetic precursors of the xanthone core in *Penicillium* and *Aspergillus* species. Acetyl-CoA is the starter compound cyclized to form anthraquinone emodin (6-methyl-1,3,8-trihydroxyanthraquinone), which in turn produces chrysophanol as a result of 6-deoxygenation. The quinone ring of chrysophanol is cleaved by enzymes MdpL (Baeyer–Villiger oxidase) and MdpJ (glutathione S transferase) leading to the formation of thioester intermediate, which is in turn reduced by the action of an oxidoreductase MdpK to benzophenone alcohol. The latter compound is dehydrated producing 1-hydroxy-6-methyl-8-hydroxymethylxanthone which is further hydroxylated to 1,7-dihydroxy-6-methyl-8-hydroxymethylxanthone by MdpD, a monooxygenase [281]. As observed in *Aspergillus nidulans*, this compound then undergoes two consecutive prenylations (O-prenylation and C-prenylation) by prenyltransferase enzymes XptB and XptA. The O-prenylation catalyzed by XptB forms variecoxanthone Awhich undergoes a C-prenylation step by XptA leading to the final product emericellin, which in turn cyclizes due to the oxidoreductase XptC to shamixanthone and epishamixanthone, as reported in *Aspergillus variecolor* and *Aspergillus rugulosus*, respectively [282,283]. Other prenylated xanthones were identified in Ascomycetes fungi as *Aspergillus* and *Penicillium* genera. In other fungi, such as *Paecilomyces variotii*, Acetyl-CoA/Malonyl-CoA are cyclized by AgnPKS to form a PKS-bound octaketide that is hydrolyzed in the atochrysone carboxylic acid by AgnL7. The atochrysone carboxylic acid is decarboxylated to emodin anthrone by AgnL1 and then oxidized to emodin by AgnL2. Emodin can be reduced by AgnL4 to dihydroquinone and then to hydroxyketone by AgnL6. AgnL8 is responsible for dehydration reactions leading to chrysophanol. The following Baeyer–Villiger oxidation carried out by monooxygenase AgnL3 forms monodictylactone, whose hydrolyzation to monodictyphenone and reduction to dihydro-monodictyphenone leads to Agnestin C and a rearrangement to either A and B, which are interconvertible [284]. The xanthone biosynthetic pathway in fungi is reported in Figure 9. In a recent review, Khattab and Farag 2022 [281] widely describe the unique structural characteristics of dimeric, dihydro-, tetrahydro, or hexahydroxanthones as well as prenylated and chlorinated xanthones in terrestrial and marine fungi. Fungi-derived lichens are reported in Table 2.

Lichens are symbiotic organisms that are composed of fungi and algae and/or cyanobacteria. They produce a variety of characteristic xanthones metabolites with various biological properties including antimicrobial, antiviral, and antitumor activities.

The biosynthesis of xanthones proceeds through the polyacetate/polymalonate pathway, where the single polyketide chain undergoes ring-closure, and possibly through a benzophenone intermediate gives two distinct series of xanthones, depending on this folding pattern. In the first pathway, the single polyketide undergoes the aldol condensation and Claisen-type cyclization to form a benzophenone intermediate that might spontaneously dehydrate to obtain the central pyrone core. This biosynthetic pathway gives rise to the common oxygen substitution pattern of lichexanthone and norlichexanthone characterized by a methyl group in the 8-position (1,3,6-trihydroxy-8-methylxanthone) [2]. A limited number of structures derived from a biosynthetically distinct pathway gives the ravenelin skeleton characterized by a methyl group in the 3-position. This biosynthetic pathway starts with the widespread anthraquinone emodin as a precursor. The cleavage of the hydroxyl group on C-6 of the emodin leads to chrysophanol, as observed in the fungus *Pyrenochaeta terrestris* [285,286]. After the oxidative ring opens, the hydroxyl group on C-4 is then incorporated and an aryl epoxidation across an A-ring edge of chrysophanol yields an intermediate which has lost its A-ring aromaticity, as proposed by Henry and Townsend [287]. This intermediate, which is stabilized by a hydrogen bond between its newly formed phenol group and the neighboring quinone group, recovers its A-ring aromaticity to grant islandicin as a shunt product. Alternatively, a second oxidation, most likely by the same P450 oxygenase, occurs to afford a Baeyer–Villiger cleavage of the central quinone ring to yield an ortho carboxybenzophenone that might follow several metabolic fates [2].

A first possibility is the 1,4-addition of a B-ring phenol to the A-ring dienone followed by dehydration and decarboxylation to access ravenelin-like xanthones after a final oxidation [287]. These reactions lead to xanthones displaying an archetypical 1,4,8-trihydroxy-3-methylxanthone skeleton. A second metabolic pathway, granting access to eumitrins and secalonic acids, assumes the methylation of the carboxy group to prevent its subsequent elimination after a similar 1,4-addition. Finally, a subsequent 1,2-addition to the benzophenone intermediate leads to further cores similar to that of tajixanthone produced by *A. variecolor*, a skeleton thus far unknown from lichens. The xanthone biosynthetic pathway in lichens is reported in Figure 10.

From this point, the other compounds differ in the position and extent of substitution, including hydroxylation, methylation of these hydroxyl groups, and chlorination [2,3]. Lichen-derived xanthones are reported in Table 3.

It is interesting to underline that even though xanthones from free-living fungi are well known, an algae-fungus collaboration has been suggested for the synthesis of several lichen xanthones. As observed in lichen *Lecanora dispersa* [222] or in *Lecanora rupicola* [288], when the fungus is cultivated in the absence of the alga, the xanthone production is diverted to other secondary metabolites being produced (e.g., depsidones such as pannarin and related compounds). However, lichens offer the widest diversity of compounds in the fungal realm, even though their bioactivities remain under-investigated despite being widely considered a promising class of compounds exerting pleiotropic pharmacological activities.

## 4. Organ and Tissue Localization of Xanthones and Their Possible Functions in Plants

From the chemical studies, data shown in the literature made it possible to obtain information about the different types of plant xanthones identified in plants and which are the organs of accumulation. On the contrary, little is known about the tissue sites of xanthones biosynthesis in plants. Tissue localization of xanthone biosynthesis has been investigated in a few studies which are described below. Immunofluorescence localization of polyketide synthase key enzymes of flavonoids, and xanthone biosynthesis, namely chalcone, and benzophenone synthases were carried out in the leaves [289] of *H. perforatum*. Benzophenones are metabolized to xanthones through benzophenone synthase (BPS) activity [258,290]. Upon mutation in a single active site position, *H. androsaemum* BPS formed phenylpyrones [291]. Berkleir and collaborators [289] studied cross-sectioned leaves of *H. perforatum* incubated with anti-His6–BPS IgG and anti–His6-CHS IgG at various developmental stages. Immunofluorescence localization of both CHS and BPS was in the mesophyll and the intensity of immunofluorescence varied with leaf age. Maximum immunolabeling of CHS was observed in approximately 0.5 cm long leaves, while BPS was undetectable. The CHS-specific fluorescence rapidly decreased in more developed leaves (1 cm long), which instead presented high levels of BPS immunofluorescence. Unlike leaves, the roots appear to be the richest organs in xanthones [15,178,292], which is consistent with the high level of BPS transcription found in both *H. sampsonii* and *H. perforatum* roots [172,178]. In situ detection of BPS transcripts and proteins has also been carried out in situ mRNA hybridization and indirect immunofluorescence detection. Moreover, label-free localization of xanthones was studied by AP-SMALDI-FT/MS imaging [172]. It should also be noted that the concentration of xanthone precursors, particularly polyprenylated benzophenones, is very high in the root system [172]. BPS protein was immunodetected in the root exodermis and the endodermis but not in the epidermis. The exodermis and the endodermis are the outermost and innermost layers of the root cortex. The authors emphasize that these tissues are structurally and functionally related [293,294], sharing the barrier role and controlling the radial transport of water and solutes in the root. As is well known, both tissues also play a role in defense against pathogens. Tocci and co-workers [22] demonstrated that root cultures of *H. perforatum* treated with the elicitor chitosan, which mimics the fungal pathogen attack, showed high levels of xanthone content. In a subsequent study performed through the 1H-NMR-based metabolomics approach, it has been observed that *H. perforatum* root cultures elicited by chitosan, and under “overcrowding stress”, produced a yield of total xanthones ten times higher compared to the previous study. Moreover, in this study the brasilixanthone B has been isolated and identified in *H. perforatum* for the first time [295]. Strengthening the defensive role of xanthones, Huang and collaborators in *H. sampsonii* [178,296] showed that cDNAs encoding HsBPS and HsCHS were differentially regulated in the vegetative and in reproductive stages. In the vegetative stage, HsBPS was highly expressed in the roots; its transcript level was approximately 100 times higher than that of HsCHS, whereas the young leaves contained higher transcript levels of HsCHS. In the reproductive stage, maximum HsCHS expression was detected in flowers, the transcript level being approximately five times higher than that of HsBPS. The inverted situation with a 10-fold difference in the expression levels was observed in the fruits.

To prove the defensive role of xanthones against fungal pathogens, Crockett and co-workers [297] demonstrated that 1,6-dihydroxy-5-methoxy-4′,5′-dihydro-4′,4′,5′-trimethylfurano-(2′,3′:3,4)-xanthone, isolated by *H. perforatum* roots, inhibited the plant pathogenic fungi *Phomopsis obscurans* and *P. viticola*.

As shown in Table 1, several papers are published on xanthone production in other genera and species belonging to other families besides the Hypericaeae; however, organ and tissue localization studies are few.

Xanthones in *Calophyllum inophyllum* L. (Calophyllaceae) roots have been studied [298]. Different xanthones were accumulated in root bark and in root hearthwood. A new xanthone named caloxanthone D has been found in the bark and caloxanthone E in the hearthwood. Chemical investigation of dichloromethane and ethyl acetate extracts from the stem and root bark of *Trema orientalis* L. (Ulmaceae) led to the isolation of 16 compounds, including four xanthones [101].

Xanthones have also recently been found in the flowers of Japanese *Gentiana* cultivars, which show red petals rather than blue. The authors characterized the pigments responsible for the red color in these cultivars revealing the presence of cyanidin-based anthocyanins and xanthones. In particular, two compounds have been identified for the xanthones: norathyriol 6-O-glucoside and norathyriol-6-O-(60 -O-malonyl)-glucosid. These compounds contributed to the red color of flowers [33].

Xanthone content has also been studied in the leaves of *Coffea pseudozanguebariae* Bridson (family Rubiaceae) wild-grown plants, collected at different developmental stages [197]. The authors showed that C-glycosylated xanthones, i.e., mangiferin 1 and isomangiferin 2, represented 6% of the dry mass in the young leaves, while they were much lower in the older leaves; these results support the hypothesis that the xanthones play a defense role in the most delicate phases of the leaf development, as demonstrated in *Hypericum*. In a subsequent study, Talamond and co-workers [32] using a multiphoton fluorescence imaging, demonstrated that mangiferin, identified as the spectrum emission, was localized in the upper epidermis and in some mesophyll cells.

Still in *Coffea*, leaf phenolic composition has been studied in 23 species and focus on mangiferin content [299]. Leaves of *Coffea arabica* L., *Coffea canephora* Pierre ex A. Froehner, *Coffea eugenioides* S. Moore and *C. pseudozanguebariae* were sampled at different developmental stages. In particular, leaves were collected at three stages: (1) young leaves from the apex; (2) leaves from the first node below the apex; (3) leaves from the second node below the apex. Microscopic observations carried out with UV light (filter UV-1A: 365 nm excitation filter) revealed yellow autofluorescence of mangiferin and its preferential localization in palisade and spongy parenchyma tissues of *C. pseudozanguebariae* leaves. In contrast, mangiferin was absent in *C. canephora* and present at low concentration in *C. arabica*. In the same study, xanthone localization in the fruits has been studied. Samples were obtained from *C. pseudozanguebariae* and *C. canephora* and collected at three developmental stages: (1) fruit when green with a partially formed seed (immature); (2) fruit when yellowish green, pericarp (exocarp, mesocarp, endocarp) and (3) fruit when reddish yellow. An intense yellow autofluorescence was shown in the cells of the exocarp and the external layers of the mesocarp of young green fruits, indicating high content of mangiferin. Mangiferin was absent in the seeds and endocarp of the three species examined. The authors speculate that the mangiferin accumulation within the fruit could be associated with photosynthetic tissues; in fact, the receptacle and young fruit are green and photosynthetic, while the ovary is not.

Moreover, in the same study, Campa and collaborators [299] demonstrated that mangiferin is accumulated in the leaves and fruits of seven of 23 *Coffea* species (24 taxa) studied, including two hybrids (*C. arabica* ‘Laurina’ and C. heterocalyx Stoff. cf.), originating from different localities in Africa. On the contrary, none of the Madagascan species contained mangiferin, perhaps due to the different environmental conditions. A relationship between mangiferin accumulation, altitude, and UV levels was speculated. The peripheral localization of mangiferin in *Coffea* plant organs and its association with photosynthetic tissue strengthens the hypothesis of a protective action against UV-radiation. Thus, in addition to a defense role against pathogens, xanthones also appear to play a role toward environmental factors. Soil conditions and altitude influenced xanthone content of *H. perforatum* roots. Young wild plants of *H. perforatum* subsp.angustifolium collected in two areas (Lazio Region, Italy) at two different altitudes (68 and 453 m above sea level) and in different soils (calcareous and volcanic) showed different amounts and quality of xanthones [13].

Some authors [242,300,301] consider xanthones a powerful antioxidant system and that they effectively suppress ROS production and prevent lipid peroxidation. Moreover, they could play a role in adaptation to environmental change [302].

## 5. Recent Insight on Biological Activities of Xanthones

The ability of xanthones to bind to multiple and unrelated classes of protein receptors as high affinity ligands, allow these molecules to be considered “promiscuous binders”, because they are able to interfere with a variety of biological targets exerting pronounced pharmacological activity against several diseases [17]. This ability is related with some special molecular features, such as the presence of the heteroaromatic tricyclic ring system being predominantly planar and rigid, the carbonyl group at the central ring capable of several interactions, the biaryl ether group contributing to the electronic system, and the xanthone core that accommodates a vast variety of substituents at different positions. Their interesting structural scaffold and pharmacological importance have encouraged scientists to isolate these compounds from natural products and synthesize them as novel drug candidates in the field of medicinal chemistry. Numerous naturally occurring and synthetic xanthone derivatives have been reported in the literature with several beneficial heterogeneous pharmacological activities. According to several authors, anticancer, antimicrobial, antimalarial, anti-HIV, anticonvulsant, anticholinesterase, antioxidant, anti-inflammatory, as well as an inhibitory activity on different enzymes, including a-glucosidase, topoisomerase, protein kinase C, miRNA, intestinal P-glycoprotein, acyl-CoA:cholesterol acyltransferase, xanthine oxidase, and aromatase have been attributed to xanthones [16,30]. All xanthone classes have already demonstrated cytotoxic effects. An antiproliferative activity of xanthone carbaldehyde derivatives, prenylated xanthones, and chiral xanthone derivatives has been demonstrated in MCF-7 (breast adenocarcinoma), KB 3.1 (squamous cell oral carcinoma), A375-C5 (melanoma), and NCI-H460 (non-small cell lung cancer) cell lines [303,304,305]. The growth inhibitory effect on human tumor cell lines was dependent on the nature and position of substituents on the xanthone scaffold and the stereochemistry of the xanthones. Among them, the major group of naturally occurring xanthones are prenylated xanthones, in which the presence of prenyl groups in key positions on the xanthone nucleus can influence the physicochemical properties, namely lipophilicity, and affect the interaction with the biological targets exerting several biological activities, such as antitumor, anti-inflammatory, and human lymphocyte proliferation inhibitory effects [306]. In this context, C-prenylated xanthones are able to decrease cellular proliferation and induce S-phase cell cycle arrest and apoptosis, increasing cleaved PARP and Bid levels and decreasing Bcl-xL in K-562 cells [307] in MCF-7 (breast adenocarcinoma), NCI-H460 (non-small cell lung cancer), A375- C5 (melanoma), and HL-60 (acute myeloid leukemia) cell lines [30]. It is known that the p53 tumor suppressor is a major transcription factor with a crucial role in cell proliferation and death. The activity of p53 is commonly lost in cancers either by mutation in the TP53 gene, or by inactivation due to the overexpression of the main endogenous negative regulator, murine double minute 2 (MDM2). Therefore, restoration of p53 activity by inhibiting the MDM2-p53 interaction represents an appealing therapeutic strategy for many wild-type p53 tumors with overexpressed MDM2 [308]. It has been demonstrated that prenylated xanthone α-mangostin and gambogic acid are inhibitors of MDM2-p53 interaction [309], while the pyranoxanthone has shown a promising growth inhibitory activity as a putative inhibitor of MDM2-p53 interaction in human tumor cells expressing wild-type p53 and overexpressed MDM2 [308]. Moreover, the oxygenated xanthones characterized by simple substituents such as hydroxyl, methoxy, or methyl groups showed antioxidant properties implicating cancer chemopreventive [310], hepatoprotective [311], antifungal [312], antibacterial [313], and anti-obesity [314] actions involving targets such as monoaminoxidase (MAO) [315], P-glycoprotein (P-gp) [316], protein kinase C (PKC) [317], and tyrosinase [318]. Concerning MAO studies, xanthones acted preferentially as MAO-A competitive, reversible inhibitors with IC50 values in the micro- to nanomolar range, and 1,5- dihydroxy-3-methoxyxanthone with an IC50 of 40 nM for MAO-A emerged as the most active inhibitor. Along with xanthonolignoids, 3,4-dihydroxy xanthone derivatives with synthetic intermediates of 3,4-dihydroxy-2-methoxyxanthone and 2,3-dihydroxy-4-methoxyxanthone were found to be the most potent lignoids, with promising antiproliferative and apoptotic effects in leukemia cell lines [30,307]. 1,2-dihydroxyxanthone, initially considered promising for its effect against melanoma [319], due to its catechol structure peri to carbonyl, is also the most promising antioxidant agent for its chelating properties, stability, phototoxicity, cytotoxic effect on a human keratinocyte cell line [320], and its modulatory effects on the activity of the THP-1 macrophage cell line, namely cytokine production [321]. Rosa et al. [318] found that the partial negative surface area, the relative number of oxygen atoms, and the substitution pattern of the 1-methyl-3,4,6-trihydroxyxanthone contributed to the tyrosinase inhibitory activity. Methoxylated xanthone derivatives were found to be promising PKC activators showing high selectivity for individual PKC isoforms, proving their utility for a detailed study of the physiological and pathophysiological roles of PKC isoforms [317,322,323]. Among the most promising xanthones for activities in which redox mechanisms are involved, it is interesting to mention dihydroxyxanthones, particularly those with a catechol moiety considered PAINS, or pan-assay interference compounds [324]. Although their activity does not depend on a specific, drug-like interaction between the molecule and a protein, dihydroxyxanthones are able to coat a protein or sequester metal ions that are essential to a protein’s function. These mechanisms are recognized for some FDA approved-drugs. Several authors have highlighted the antimicrobial activity of xanthones against diverse human pathogenic microorganisms. The antimicrobial activity of synthetic xanthones, xanthenediones, and spirobenzofurans against the yeasts *Cryptococcus neoformans* and *Candida albicans* has been reported [22,325,326]. Hydroxyxanthones have been proposed as novel antimalarial agents, with activity against multidrug-resistant Plasmodium parasites being able to exert complexation to the heme and inhibition of hemozoin formation [327]. Interestingly, 1,3-dihydroxyxanthone derivatives showed the ability to inhibit acetylcholinesterase (AChE) and block the acetylcholinesterase-induced by-amyloid aggregation [70]. The research for new cholinesterase inhibitors is an important strategy to identify new drug candidates to treat Alzheimer’s disease and related dementias. Most currently known natural inhibitors of acetylcholinesterase (AChE) are alkaloids, which have the disadvantages of short half-lives and/or undesirable side effects [328]. A pool of xanthones, such as bellidifolin, bellidin, swertianolin, and norswertianolin from *Gentiana campestris* Geners. exhibited potent inhibitory activities against AChE with MIC values of 0.01, 0.04, 0.08, and 0.5 μM, respectively [329]. Reutrakul et al. [330,331] demonstrated the anti-HIV-1 activities of the 1,3,8-trihydroxy-2,4-dimethoxyxanthone and euxanthone from *Cratoxylum arborescens* Blume in the syncytium assay, with EC50 values of 17.9 and 18.8 μM [330]. In addition, morellic acid, gambogic acid, and dihydroisomorellin have shown moderate HIV-1 inhibitory activities in the reverse transcriptase assay, with IC50 values of 11, 15, and 42.3 μM, respectively [331]. Xanthones have been shown to have beneficial effects on several cardiovascular diseases, including atherosclerosis, hypertension, thrombosis and ischemic heart disease [332]. Wang et al. [333] have shown that 1-hydroxy 2,3,5-trimethoxyxanthone, a tetraoxygenated xanthone from *Halenia elliptica* D.don (Gentianaceae), induces potent concentration-dependent relaxation in rat coronary artery rings pre-contracted with 1 μM of 5- hydroxytryptamine (EC50, 1.67 μM), while one of its major metabolites, 1,5-dihydroxy-2,3-dimethoxyxanthone, induces a relaxation effect with an EC50 of 4.4 μM. It is important to underline that single xanthones may have multiple pharmacological effects, since pharmacophores with diverse effects share the same tricyclic scaffold but differ in the nature and/or positions of substituents. It should be noted that the inventory of natural xanthones remains far from complete, and the functional-group diversity and architectural platforms of natural products generated in their biosynthesis continue to provide new information for synthetic and medicinal chemists in strategies for making biologically active mimics.

## 6. Conclusions

This review describes the biosynthetic process of xanthone in plants, fungi, and lichens which has yet to be updated comprehensively in the last decade. In higher plants, xanthone biosynthesis involves the shikimate and the acetate pathways which originate in plastids and endoplasmic reticulum, respectively. The pathway continues following three alternative routes, two phenylalanine-dependent and one phenylalanine-independent. However, all three routes lead to the biosynthesis of 2,3′,4,6-tetrahydroxybenzophenone, which is the central intermediate of xanthone biosynthesis. Unlike plants, the xanthone core in fungi and lichens is wholly derived from polyketide. Several xanthone derivatives can be originated from these precursors and differ between plants, fungi, and lichens. Despite there being several studies on chemical and biochemical synthesis of xanthones in plants, there has been little investigation on their subcellular, cellular, and tissue trafficking. As it was reported in this review, these issues have been deeply explored only in few species, including *Hypericum* spp. and a few others. No study has been reported about these aspects in fungi and lichens. Xanthones are molecules involved in defense response to both biotic and abiotic agents in plants, although their role in fungi and lichens has not yet been exhaustively explored. Interestingly, xanthones derived from plants, fungi, and lichens show biological activities in many human diseases. In this context, further knowledge of mechanisms underlying xanthone biosynthesis in different plant organisms will be useful to optimize the production of these high-value products for application purposes.

## Figures and Tables

**Figure 1 plants-12-00694-f001:**
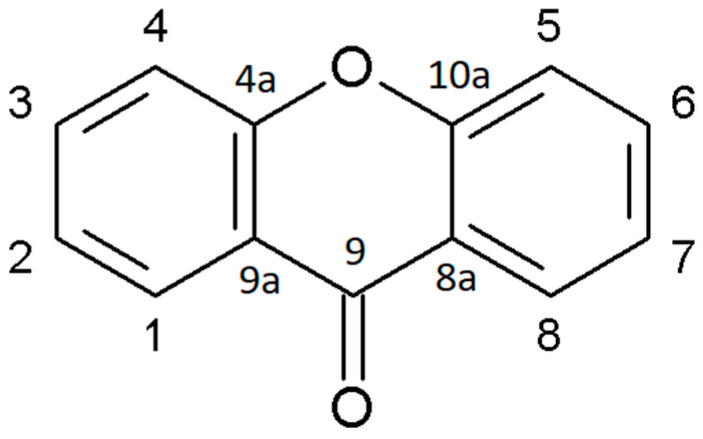
Structure of 9H-xanthen-9-one. A-ring (carbons 1–4) and B-ring (carbons 5–8) are attached through an oxygen atom and a carbonyl group.

**Figure 2 plants-12-00694-f002:**
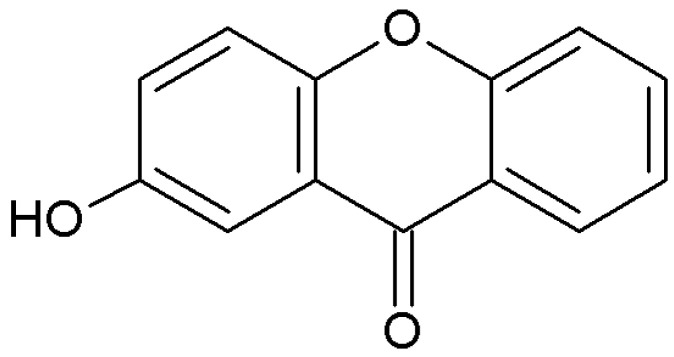
Structure of 2-hydroxyxanthone, oxygenated xanthone.

**Figure 3 plants-12-00694-f003:**
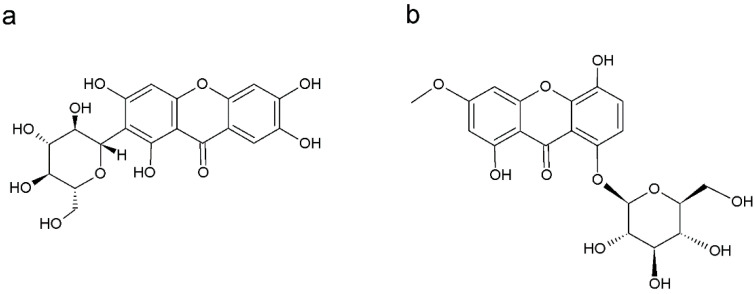
Structure of mangiferin, C-glycoside (**a**) and swertianolin, O-glycoside (**b**).

**Figure 4 plants-12-00694-f004:**
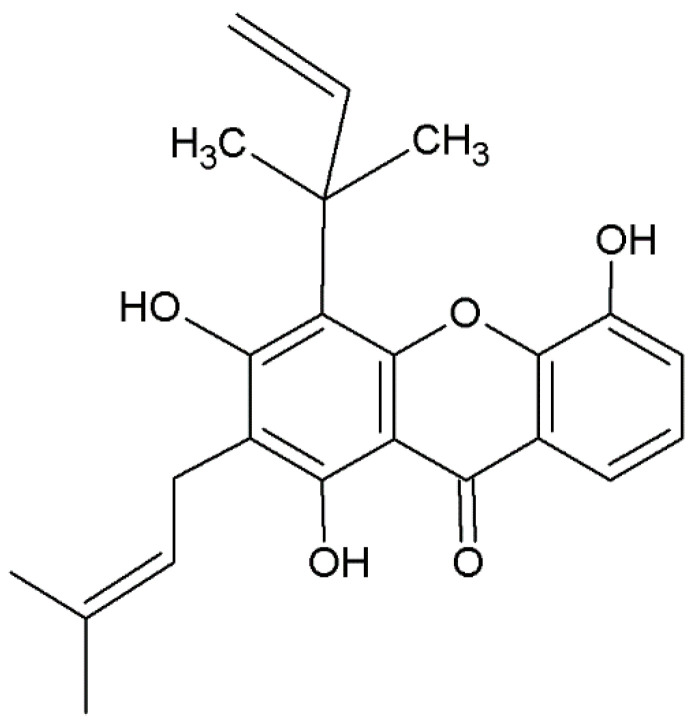
Structure of allanxanthone A, prenylated xanthone.

**Figure 5 plants-12-00694-f005:**
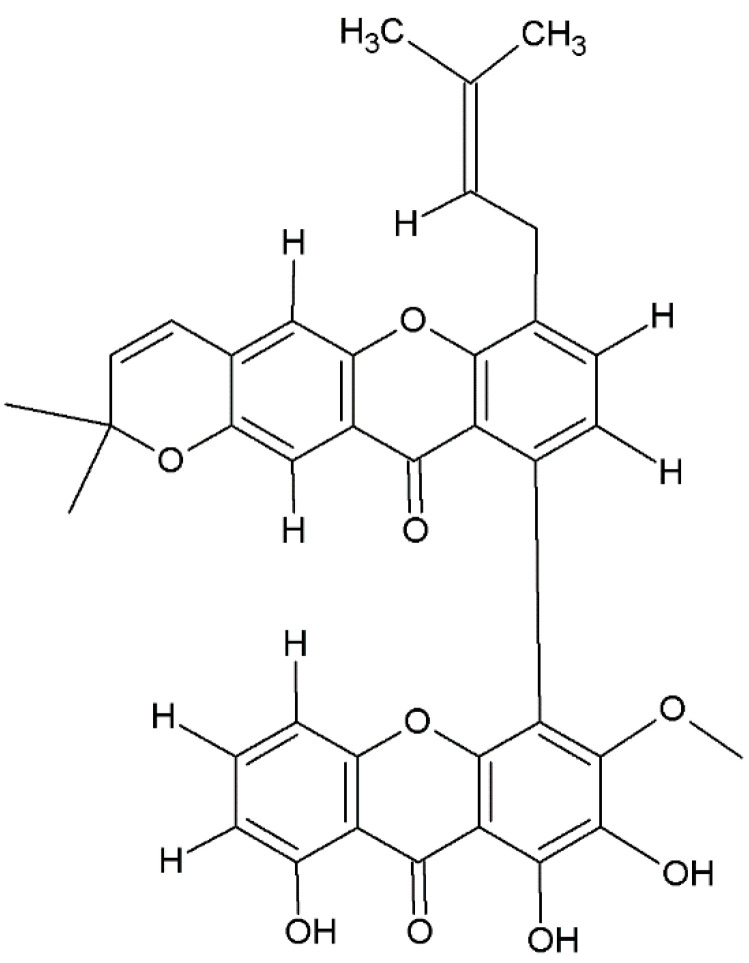
Structure of globulixanthone E, bisxanthone.

**Figure 6 plants-12-00694-f006:**
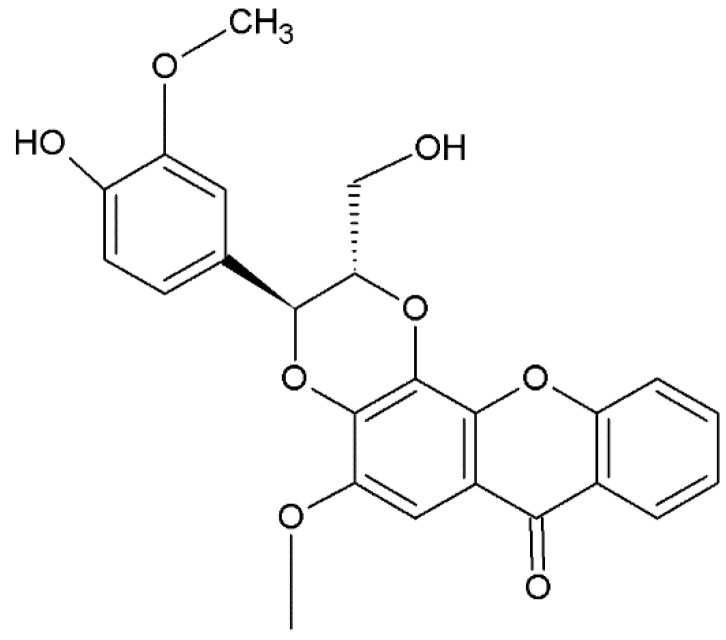
Structure of kielcorin, xantholignoid.

**Figure 7 plants-12-00694-f007:**
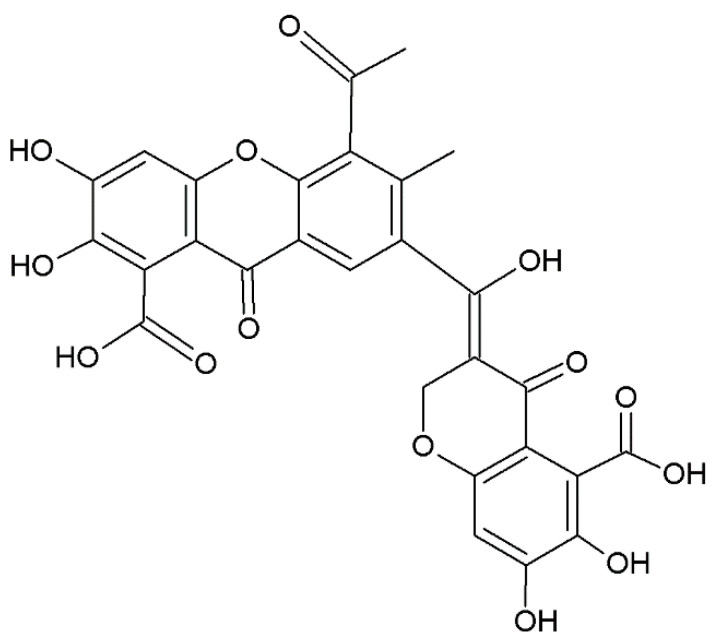
Structure of xanthofulvin, miscellaneous xanthone.

**Figure 8 plants-12-00694-f008:**
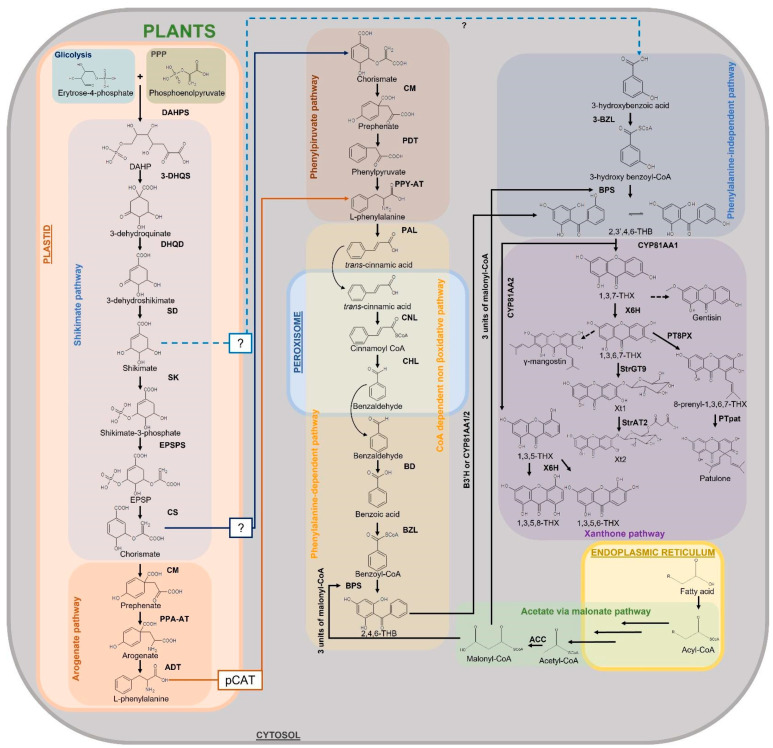
Pathways involved in xanthone biosynthesis in plants. ?: unknown proteins; 1,3,5,6-THX: 1,3,5,6-tetrahydroxyxanthone; 1,3,5,8-THX: 1,3,5,8-tetrahydroxyxanthone; 1,3,5-THX: 1,3,5-trihydroxyxanthone; 1,3,6,7-THX: 1,3,6,7-THX-tetrahydroxyxanthone; 1,3,7-THX: 1,3,7-trihydroxyxanthone; 2,3′,4,6-THB: 2,3′,4,6-tetrahydroxybenzophenone; 2,4,6-THB: 2,4,6-trihydroxybenzophenone; 3-BZL: 3-benzoate-CoA ligase; 3-DHQS: 3-dehydroquianate synthase; 8-prenyl-1,3,6,7-THX: 8-prenyl-1,3,6,7-tetrahydroxyxanthone; ACC: acetyl-CoA carboxylase; ADT: arogenate dehydratase; B3′H: benzophenone 3′-hydroxylase; BD: benzaldehyde dehydrogenase; BPS: benzophenone synthase; BZL: benzoate-CoA ligase; CHL: cinnamoyl-CoA hydratase/lyase; CM: chorismate mutase; CNL: cinnamate-CoA ligase; CoASH: coenzyme A; CS: chorismate synthase; CYP81AA1/2: Cytochrome P450 oxydase 81AA1/2; DAHP: 3-deoxy-D-arabino-heptulosonate-7-phosphate; DAHPS: DAHP synthase; DHQD: 3-dehydroquinase dehydratase; EPSP: 5-enolpyruvylshikimate 3-phosphate; EPSPS: EPSP synthase; PAL: phenylalanine ammonia lyase; pCAT: plastidial cationic amino acid transporter; PDT: prephenate dehydratase; PPA-AT: prephenate aminotransferase; PPP: pentose phosphate pathway; PPY-AT: phenylpyruvate aminotransferase; PT8PX: 8-prenylxanthone-forming prenyltransferase; PTpat: patulone-forming prenyltransferase; SD: shikimate 5-dehydrogenase; SK: shikimate kinase; StrAT2: malonyl-CoA acyltransferase; StrGT9: norathyriol 6-O-glucosyltransferase; X6H: xanthone-6-hydroxylases; Xt1: norathyriol 6-O-glucoside; Xt2: norathyriol-6-O-(6′-Omalonyl)-glucoside.

**Figure 9 plants-12-00694-f009:**
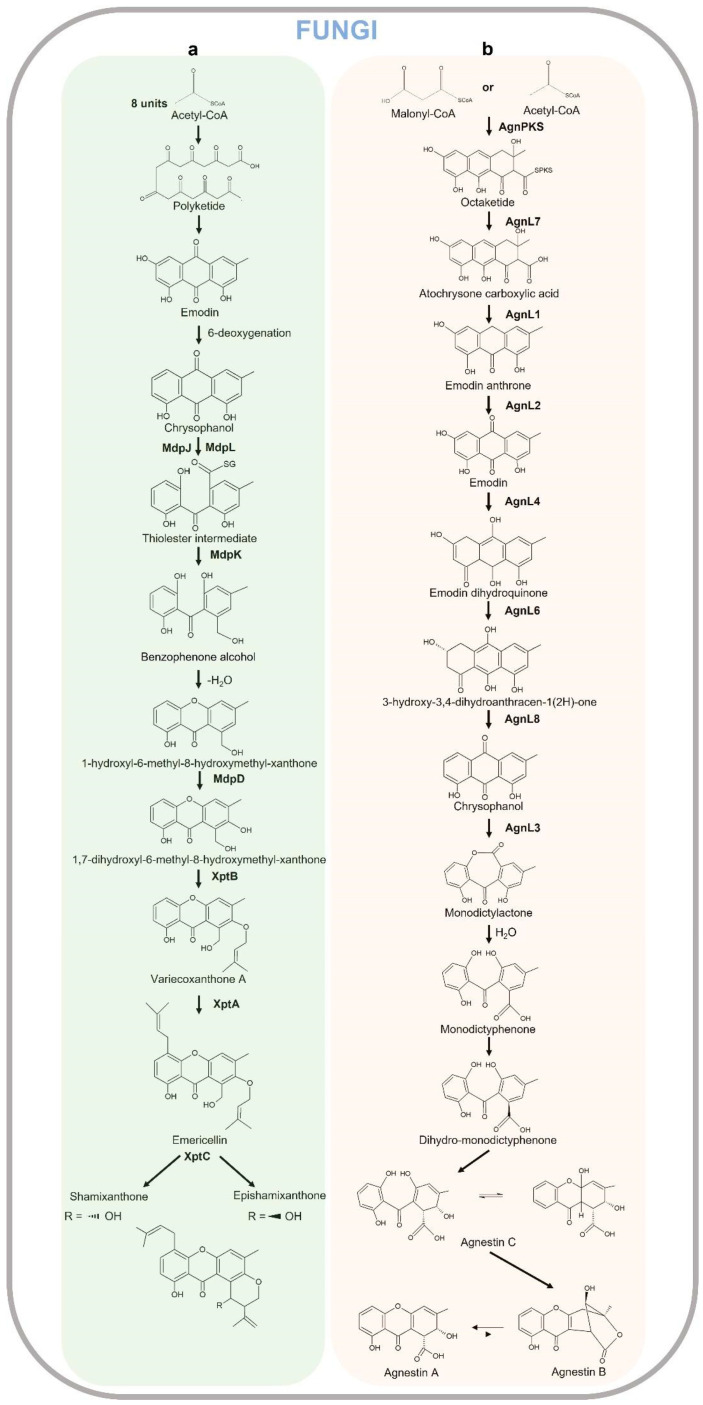
Xanthone biosynthetic pathways in fungi. Shamixanthone (**a**) and agnestin pathway (**b**).

**Figure 10 plants-12-00694-f010:**
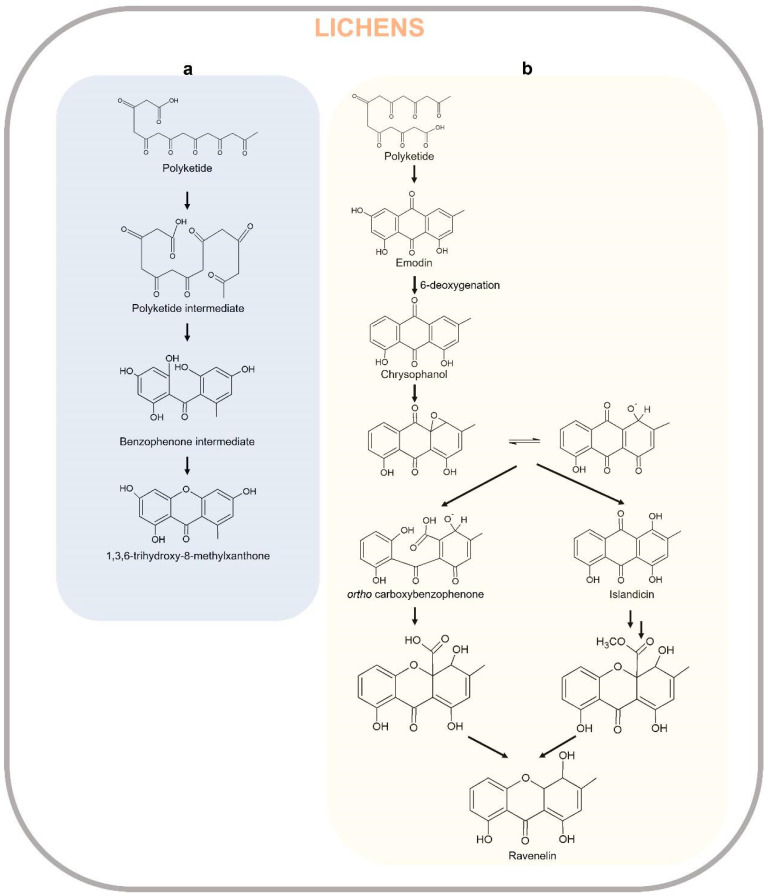
Xanthone biosynthetic pathways in lichens. Lichexanthone-type lichen xanthone (**a**) and thiomielin-type lichen xanthone pathway (**b**).

**Table 1 plants-12-00694-t001:** Xanthones in Plants.

Plants	Xanthone	Organ/Tissue	References
Acanthaceae	Oxygenated xanthones	Root	
*Andrographis paniculata* (Burm.f.)Nees			[34,35]
Anacardiaceae	Xanthone glycosides	Fruit peel	
*Mangifera indica* L.			[36,37]
Annonaceae	Oxygenated xanthones Prenylated xanthones Xantholignoids	Fruit, resin, leaf, heartwood	
*Anaxagorea luzonensis* A.Gray			[38]
*Orophea corymbosa* Miq.			[39]
Asparagaceae	Xanthone glycosides, oxygenated xanthones Xantholignoids	Rootstock, apical part, bulb, tuber	
*Anemarrhena asphodeloides* Bunge			[40,41,42]
*Drimiopsis maculata* Lindl. & Paxton			[43]
*Ledebouria graminifolia* (Baker) Jessop			[44]
Asteraceae	Oxygenated xanthones Xantholignoids	Leaf	
*Senecio mikanioides* Otto ex Harv.			[45]
Bignoniaceae	Xanthone glycosides	Apical part	
*Arrabidaea samydoides* (Cham.) Sandwith			[46]
Bombacaceae	Xanthone glycosides	Leaf, flower	
*Bombax ceiba* L.			[47,48]
*B. malabaricum* DC.			[49]
Bonnetiaceae	Xanthone glycosides	Apical part	
*Bonnetia dinizii* Huber			[50]
Calophyllaceae *Calophyllum apetalum* Willd.	Oxygenated xanthones Xantholignoids	Heartwood stem bark, seed, root, wood	[51]
*C. austroindicum* Kosterm. ex P.F.Stevens			[52]
*C. bracteatum* Thwaites			[53]
*C. brasiliense* Vesque			[53,54,55,56,57,58,59,60]
*C. calaba* L.			[53,61]
*C. canum* Hook.f. ex T.Anderson			[62]
*C. caledonicum* Vieill. ex Planch. & Triana			[63,64,65,66,67]
*C. castaneum* P.F.Stevens			[9]
*C. fragrans* Ridl.			[68]
*C. inophyllum* L.			[69,70,71,72,73,74,75,76,77,78]
*C. moonii* Wight			[79]
*C. neo-ebudicum* Guillaumin			[80]
*C. ramiflorum* O.Schwarz			[81]
*C. sclerophyllum* Vesque			[82]
*C. scriblitifolium* M.R.Hend. & Wyatt-Sm.			[83]
*C. tetrapterum* Miq.			[84]
*C. tomentosum* Wight			[85,86]
*C. zeylanicum* Kosterm.			[87]
*Caraipa densiflora* Kubitzki			[88,89]
*Haploclathra leiantha* Benth.			[90,91]
*H. paniculata* Benth.			[92,93]
*Kielmeyera coriacea* Mart. & Zucc.			[88]
*K. ferruginea* A.P.B.Santos & Trad			[94]
*K. rupestris* Duarte			[95,96,97]
*K. speciosa* A.St.-Hil., A.Juss. & Cambess.			[98]
*K. variabilis* Mart. & Zucc.			[99]
*Mesua ferrea* L.			[78,100]
Cannabaceae	Bisxanthones	Bark	
*Trema orientalis* (L.) Blume			[101]
Caryophyllaceae	Oxygenated xanthones	Apical part	
*Saponaria vaccaria* L.			[102,103]
Celastraceae	Xanthone glycoside	Root	
*Salacia reticulata* Wight			[104]
Clusiaceae	Prenylated xanthones Xantholignoids Bisxanthones Oxygenated xanthones	Heartwood, stem bark, fruit, seeds, leaf, root	
*Allanblackia floribunda* Oliv.			[105,106]
*A. monticola* Mildbr. ex Engl.			[107,108]
*Garcinia cowa* Roxb.			[109,110]
*G. echinocarpa* Thwaites			[111]
*G. forbesii* King			[112]
*G. mangostana* L.			[113,114,115,116,117,118,119,120,121,122,123]
*G. nobilis* Engl.			[124]
*G. opaca* King			[125]
*G. ovalifolia* Oliv.			[126]
*G. paucinervis* Chun & F.C.How			[117]
*G. pedunculata* Roxb. ex Buch.-Ham.			[127]
*G. quadrifaria* Baill. ex Pierre			[128]
*G. staudtii* Engl.			[128]
*G. terpnophylla* Thwaites			[113]
*G. vieillardii* Pierre			[129]
*G. xanthochymus* Hook.f.			[130]
*Pentadesma butyracea* Sabine			[131]
*Symphonia globulifera* L.f.			[132,133]
Eriocaulaceae	Oxygenated xanthones	Apical part	
*Leiothrix curvifolia* (Bong.) Ruhland			[134]
*L. flavescens* (Bong.) Ruhland			[134]
Fabaceae	Xanthone glycosides Prenylated xanthones	Shoot	
*Baphia kirkii* Baker			[135]
*Cyclopia genistoides* (L.) R.Br.			[136]
*C. intermedia* E.Mey.			[136]
*C. maculata* (Andrews) Kies			[136]
*C. sessiliflora* Eckl. & Zeyh.			[136]
Gentianaceae	Oxygenated xanthones Xanthone glycosides	Seed, root, leaf, rhizome	
*Canscora decussata* Schult.			[78]
*Centaurium erythraea* Raf.			[137,138,139]
*C. cachanlahuen* B.L.Rob.			[140]
*C. linarifolium* (Lamark) G. Beck *Frasera caroliniensis* Walter			[78,141] [142]
*Gentiana acaulis* L.			[143]
*G. lutea* L. *G. rhodantha* Franch.			[144,145,146] [147]
*Gentianella turkestanorum* (Gand.) Holub *Hoppea fastigiata* Griseb.			[148] [149]
*Swertia chirata* Buch.-Ham. ex Wall.			[150,151]
*S. mileensis* T.N.Ho & W.L.Shih			[152]
*S. punicea* Hemsl.			[153,154]
*S. purpurascens* Wall.			[155]
*S. randaiensis* Hayata			[156]
*S. japonica* Makino			[156]
*S. swertopsis* Makino			[156]
*S. davidii* Franch.			[154,157]
Hypericaceae	Prenylated xanthones Oxygenated xanthones Xantholignoids Miscellaneous xanthones	Fruit, stem bark, root, leaf	
*Cratoxylum cochinchinense* (Lour.) Blume			[158,159]
*C. formosum* (Jack) Benth. & Hook.f. ex Dyer			[117,159,160]
*C. pruniflorum* Kurz			[161]
*Harungana madagascariensis* Lam. ex Poir.			[71]
*Hypericum androsaemum* L.			[162]
*H. canariensis* L.			[163]
*H. geminiflorum* Hemsl. *H. japonicum* Thunb			[164] [165]
*H. maculatum* Crantz			[166]
*H. oblongifolium* Choisy *H. patulum* Thunb.			[167] [168,169,170,171]
*H. perforatum* L.			[13,172]
*H. reflexum* L.f.			[173]
*H. riparium* A.Chev.			[174]
*H. roeperianum* G.W.Schimp. ex A.Rich. *H. sampsonii* Hance			[175] [176,177,178]
*H. subalatum* Hayata			[179]
*Psorospermum adamauense* Engl. *Vismia guaramirangae* Huber			[180] [181]
Iridaceae	Xanthone glycosides	Apical part	
*Iris adriatica* Trinajstić ex Mitić			[182]
*I. albicans* Lange			[183]
*I. florentina* L.			[184]
*I. germanica* L.			[185]
*I. nigricans* Dinsm.			[186]
Lamiaceae	Oxygenated xanthones	Roots	
*Premna microphylla* Turcz.			[187]
Lecythidaceae	Oxygenated xanthones	Wood, bark	
*Gustavia hexapetala* (Aubl.) Sm.			[188]
Moraceae	Oxygenated xanthones Prenylated xanthones Xantholignoids Xanthone glycoside	Root, twig, bark	
*Cudrania cochinchinensis* (Lour.) Yakuro Kudo & Masam. *Monnina obtusifolia* Kunth *Tovomita brasiliensis* Mart.			[189] [78,117] [190]
Nyssaceae	Oxygenated xanthones	Flower, fruit, stem, leaf	
*Camptotheca acuminata* Decne.			[191]
Poaceae	Xantholignoids	Apical part	
*Chionochloa flavicans* Zotov			[192]
Polygalaceae *Polygala caudata* Rehder & E.H.Wilson	Oxygenated xanthones	Root	[193]
*P. sibirica* L.			[194]
*P. tenuifolia* Willd.			[195]
*P. vulgaris* L.			[196]
Polypodiaceae	Xanthone glycoside	Whole plant	
*Pyrrosia sheareri* (Baker) Ching			[154]
Rubiaceae	Xanthone glycoside	Leaf	
*Coffea pseudozanguebariae* Bridson			[32,197]
Thymeleaceae	Xanthone glycoside	Leaf, stem	
*Gnidia involucrata* Steud. ex A.Rich.			[198]
Zingiberaceae	Prenylated xanthones	Rhizome	
*Hedychium gardnerianum* Sheppard ex Ker Gawl.			[199]

**Table 2 plants-12-00694-t002:** Xanthones in Fungi.

Fungi (Genus)	Xanthones	References
*Actinomadura*	Oxygenated xanthones Prenylated xanthones	[200,201]
*Apiospora*	Oxygenated xanthones	[202]
*Aspergillus*	Oxygenated xanthones	[3,203,204,205,206,207,208,209,210,211]
*Chaetomium*	Miscellaneous xanthones	[204]
*Emericella*	Oxygenated xanthones Prenylated xanthones Miscellaneous xanthones	[204,208,212]
*Gibberella*	Oxygenated xanthones	[213]
*Guanomyces*	Oxygenated xanthones	[214]
*Humicola*	Oxygenated xanthones	[215]
*Monodictys*	Oxygenated xanthones Miscellaneous xanthones	[204]
*Paecilomyces*	Prenylated xanthones	[204]
*Penicillium*	Oxygenated xanthones Miscellaneous xanthones	[204,205,216]
*Phomopsis*	Oxygenated xanthones	[217,218]
*Phoma*	Oxygenated xanthones	[204]
*Wardomyces*	Oxygenated xanthones Miscellaneous xanthones	[204,219]
*Xylaria*	Oxygenated xanthones Miscellaneous xanthones	[219]

**Table 3 plants-12-00694-t003:** Xanthones in Lichens.

Lichens (Genus)	Xanthones	References
*Calopadia*	Miscellaneous xanthones	[220]
*Diploicia*	Bixanthones	[221]
*Lecanora*	Oxygenated xanthones Miscellaneous xanthones	[209,222,223,224]
*Lecidella*	Oxygenated xanthones	[224]
*Myriolecis*	Oxygenated xanthones Miscellaneous xanthones	[223]
*Micarea*	Oxygenated xanthones	[10,209,224,225]
*Pertusaria*	Miscellaneous xanthones	[226]
*Phyllopsora*	Miscellaneous xanthones	[227]
*Pseudoparmelia*	Oxygenated xanthones Miscellaneous xanthones	[228]
*Pyrenula*	Oxygenated xanthones	[229]
*Sporopodium*	Miscellaneous xanthones	[230]
*Teloschistale*	Bixanthones	[2]
*Umbilicaria*	Xanthone glycosides	[231]

## Data Availability

Not applicable.

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
