# Peer review of "Xanthones: Biosynthesis and Trafficking in Plants, Fungi and Lichens"

_plants, 2023, doi:10.3390/plants12040694_

Round 1

Reviewer 1 Report

Over all the quality of the submitted manuscript is interesting. It deals with the advances in the field of Xanthones: biosynthesis and trafficking in plant organisms. In my opinion, the work is well done, and I recommend it for publication.

Nevertheless, I think that there are minor aspects that need to be improved:

Figure 1. All carbons must be numbered.

Page 2, rows 85-86, methyl is not an oxygenated substituent.

Figure 2. This figure needs to be improved. The schema looks good, but it is not easily readable. It is a standard of too much small words and structures.

Table 2. There are several problems in the reference column.

The Botanical authority of the plants is to be included.

I suggest to add at least one structure of each class of xanthones

I suggest to include a paragraph regarding the main bioactivity of these molecules

I suggest to add at least one structure of each class of xanthones

I suggest to  add a list of abbreviations

Author Response

We kindly acknowledge the editor and the reviewers for their remarkable comments and precious time dedicated to review the manuscript. Below we have answered all the questions risen by reviewers indicating also the modifications and integrations in the manuscript. We hope that you may find our answer exhaustive and that the paper, in its current form, is acceptable for publication in Plants.

Reviewer #1

Comments and suggestions for authors:

“Over all the quality of the submitted manuscript is interesting. It deals with the advances in the field of Xanthones: biosynthesis and trafficking in plant organisms. In my opinion, the work is well done, and I recommend it for publication.

Nevertheless, I think that there are minor aspects that need to be improved.”

  • “Figure 1. All carbons must be numbered.”.”

Reply: We thank the reviewer, and we modified the figure.

  • “Page 2, rows 85-86, methyl is not an oxygenated substituent.”

Reply: we apologize, this is a typo from a former version, we modified the text including this part about the methyl group to lines 77-78.

  • “Figure 2. This figure needs to be improved. The schema looks good, but it is not easily readable. It is a standard of too much small words and structures.”

Reply: We thank the reviewer. We increased the size of the text and structures in Figure 2 as much as possible respecting the criteria of the journal, the page space available and keeping the maximum resolution. 

  •  “Table 2. There are several problems in the reference column”

Reply: We apologize, we checked all the references and corrected the mistakes. 

  • “The Botanical authority of the plants is to be included.”

Reply: The Botanical authority has been included for each species.

  • “I suggest to add at least one structure of each class of xanthones”

Reply: We agree with the reviewer. A sample figure for each type of xanthone has been added to make the discussion clearer. 

  • “I suggest to include a paragraph regarding the main bioactivity of these molecules”

Reply: We thank the reviewer for this suggestion. A paragraph describing the biological activity of xanthones was added at the end of the manuscript (paragraph 5). 

  • “I suggest to  add a list of abbreviations”

Reply: We agree with the reviewer, and we added a list of abbreviations in each figure to make it easier to read

Reviewer 2 Report

In this review article, the authors summarize the current knowledge of the biosynthetic pathway in plants, fungi, and lichens, as well as the tissue sites of xanthone biosynthesis in plants. The authors also discuss the possible functions of xanthones in plants. Overall, the manuscript is well-organized and written appropriately. Because xanthones are an important family of natural products with a wide variety of biological activities, this review article would be of interest to the community. There are a few minor suggestions as follows:

1. There are many "Error! Reference source not found" words in Tables 1 and 2, which need to be corrected.

2. The authors should check the chemical structures of xanthones and biosynthetic intermediates in the biosynthetic pathways. For example, the methyl group of thiolester intermediate in Figure 3 is missing, which needs to be corrected. The methyl group of the compound that is positioned below the chrysophanol in Figure 4 is also missing.

3. It might be confusing for the readers to follow the biosynthetic pathway because the orientation of compounds is not consistent. For example, the position of the methyl group of emodin is right lower, whereas the position of the methyl group of chrysophanol is left upper (Figures 3 and 4). It is better to draw the structures of compounds with the same orientation.

4. The authors should check the reference number throughout the manuscript. For example, "[112]" of "Henry and Townsend [112]" (line 359-360) should be "[111]". "[104]" of "Birch and Donovan [104]" (line 312) should be "[103]".

Author Response

We kindly acknowledge the editor and the reviewers for their remarkable comments and precious time dedicated to review the manuscript. Below we have answered all the questions risen by reviewers indicating also the modifications and integrations in the manuscript. We hope that you may find our answer exhaustive and that the paper, in its current form, is acceptable for publication in Plants.

Reviewer #2

Comments and suggestions for authors:

“In this review article, the authors summarize the current knowledge of the biosynthetic pathway in plants, fungi, and lichens, as well as the tissue sites of xanthone biosynthesis in plants. The authors also discuss the possible functions of xanthones in plants. Overall, the manuscript is well-organized and written appropriately. Because xanthones are an important family of natural products with a wide variety of biological activities, this review article would be of interest to the community. There are a few minor suggestions as follows:” 

  • “There are many "Error! Reference source not found" words in Tables 1 and 2, which need to be corrected.”

Reply: We apologize, we checked all the references and corrected the mistakes. 

  • “The authors should check the chemical structures of xanthones and biosynthetic intermediates in the biosynthetic pathways. For example, the methyl group of thiolester intermediate in Figure 3 is missing, which needs to be corrected. The methyl group of the compound that is positioned below the chrysophanol in Figure 4 is also missing.”

Reply: We apologize for the error, all the chemical structures have been checked and corrected where it was necessary.

  • “It might be confusing for the readers to follow the biosynthetic pathway because the orientation of compounds is not consistent. For example, the position of the methyl group of emodin is right lower, whereas the position of the methyl group of chrysophanol is left upper (Figures 3 and 4). It is better to draw the structures of compounds with the same orientation.”

Reply: We agree with the reviewer, and we draw the structures with the same orientation.

  • “The authors should check the reference number throughout the manuscript. For example, "[112]" of "Henry and Townsend [112]" (line 359-360) should be "[111]". "[104]" of "Birch and Donovan [104]" (line 312) should be "[103]".”

Reply: Reference numbering has been revised and corrected when necessary.